# Piezo-Enhanced Photocatalytic Activity of the Electrospun Fibrous Magnetic PVDF/BiFeO_3_ Membrane

**DOI:** 10.3390/polym15010246

**Published:** 2023-01-03

**Authors:** Farid Orudzhev, Dinara Sobola, Shikhgasan Ramazanov, Klára Částková, Nikola Papež, Daud A. Selimov, Magomed Abdurakhmanov, Abdulatip Shuaibov, Alina Rabadanova, Rashid Gulakhmedov, Vladimír Holcman

**Affiliations:** 1Smart Materials Laboratory, Department of Inorganic Chemistry and Chemical Ecology, Dagestan State University, St. M. Gadjieva 43-a, Makhachkala 367015, Dagestan Republic, Russia; 2Amirkhanov Institute of Physics of Dagestan Federal Research Center, Russian Academy of Sciences, Makhachkala 367003, Russia; 3Department of Physics, Faculty of Electrical Engineering and Communication, Brno University of Technology, Technická 2848/8, 61600 Brno, the Czech Republic; 4Central European Institute of Technology, Purkyňova 656/123, 61200 Brno, the Czech Republic

**Keywords:** PVDF, BiFeO_3_, photocatalysis, piezocatalysis, piezo-photocatalysis, electrospinning, fibers, smart materials

## Abstract

Creating stimulus-sensitive smart catalysts capable of decomposing organic dyes with high efficiency is a critical task in ecology. Combining the advantages of photoactive piezoelectric nanomaterials and ferroelectric polymers can effectively solve this problem by collecting mechanical vibrations and light energy. Using the electrospinning method, we synthesized hybrid polymer-inorganic nanocomposite fiber membranes based on polyvinylidene fluoride (PVDF) and bismuth ferrite (BFO). The samples were studied by scanning electron microscope (SEM), Fourier-transform infrared spectroscopy (FTIR), total transmittance and diffuse reflectance, X-ray photoelectron spectroscopy (XPS), differential scanning calorimetry (DSC), thermogravimetric analysis (TGA), vibrating-sample magnetometer (VSM), and piezopotential measurements. It has been demonstrated that the addition of BFO leads to an increase in the proportion of the polar phase from 86.5% to 96.1% due to the surface ion–dipole interaction. It is shown that the composite exhibits anisotropy of magnetic properties depending on the orientation of the magnetic field. The results of piezo-photocatalytic experiments showed that under the combined action of ultrasonic treatment and irradiation with both visible and UV light, the reaction rate increased in comparison with photolysis, sonolysis, and piezocatalysis. Moreover, for PVDF/BFO, which does not exhibit photocatalytic activity, under the combined action of light and ultrasound, the reaction rate increases by about 3× under UV irradiation and by about 6× under visible light irradiation. This behavior is explained by the piezoelectric potential and the narrowing of the band gap of the composite due to mechanical stress caused by the ultrasound.

## 1. Introduction

Environmental problems associated with water pollution are the subjects of numerous studies worldwide. The processes of globalization and industrialization, combined with increasing consumption, exacerbate environmental pollution. Industrial organic dyes are some of the main mass water pollutants. It was reported that about 12% of dyes in the manufacturing process and 20% of dyes in the dyeing processes are lost and discharged with effluents [1,2]. These dye impurities have relatively high toxicity, carcinogenicity, and potentially mutagenic properties [3]. Therefore, it is critical to create methods to treat wastewater containing both cost-effective and ecologically beneficial dyes before they are released into the environment. However, the disadvantages of existing chemical methods for water purification from dyes are high energy consumption, a large number of chemicals used, and the need for special equipment [4]. Among the most efficient, environmentally friendly, less expensive, and promising approaches are advanced oxidation processes (AOPs) aimed at the decomposition and mineralization of persistent organic matter from wastewater by reaction with hydroxyl radicals (OH). AOP can be combined with ozone ($O_{3}$), catalyst, ultraviolet (UV) irradiation, ultrasound, or piezoelectric potential [5,6,7,8].

Therefore, the search for universal “smart” materials that can combine the advantages of several AOPs, as well as the properties that can be controlled through external stimuli, is relevant. One such material is ${\mathrm{BiFeO}}_{3}$ (BFO) multiferroic bismuth ferrite [9,10,11]. The presence of spontaneous polarization, magnetoelectric coupling, ferromagnetism, a narrow band gap, and the ease of control of the structure make this material very promising for use as a smart photocatalyst [12,13,14]. Numerous research studies have focused on the photocatalytic capabilities of BFO, which are summarized in review articles [15,16,17]. It has been shown in recent research studies that simultaneous exposure to light and mechanical vibrations leads to an increase in photocatalytic activity due to the piezoelectric polarization of BFO [18,19]. In work by Liu et al. [20], multiferroics (La-BFO) based on La-doped bismuth ferrite were synthesized, and it was proven that the reaction constant of the piezo-photocatalytic decomposition of carbamazepine was 2.2× higher than the sum of piezo- and photocatalysis. In [21], the authors demonstrated that the reaction constant of piezo-photocatalysis is 566% higher than that of piezocatalysis. In another research, under UV-visible irradiation and external exposure, BFO nanosheets and nanowires reduced RhB in 1 h with efficiencies of around 71% and 97%, respectively [15]. To implement effective charge separation, piezoelectric nanomaterials have been utilized to decompose organic water contaminants by mechanical vibration. Nevertheless, nanostructured piezoelectric catalysts can impact water contaminants in particulate form and limit their recyclability and reuse.

In this regard, composite materials based on polymers are particularly relevant and attractive [22]. Polymeric ferroelectrics, which are non-toxic, biocompatible, chemically inert, flexible, and can be formed into films, offer significant benefits in this area [23,24,25,26,27]. Among them, polyvinylidene fluoride (PVDF) is a great choice as a matrix for BFO since it is possible to obtain a piezo-photocatalyst similar to those previously reported [28,29]. The creation of composite polymers based on BFO for various applications was reported earlier [30,31,32,33]. However, to our knowledge, there have been no studies related to the synthesis of composite fiber membranes based on PVDF and BFO, and no investigations have been conducted to examine their photo-, piezo-, and piezo-photocatalytic properties.

Studies show that composite materials based on PVDF in the form of electrospun nanofibers have higher proportions of the piezo-active phase, crystallinity, and a high specific surface area, making them more suitable for such applications [34,35,36,37,38,39,40,41].

In this study, we synthesized a hybrid polymer-inorganic fiber membrane using electrospinning, in which BFO nanoparticles were embedded inside PVDF fibers. The morphology, structure, and magnetic properties of the composite are characterized. It has been shown that the combination of ultrasound and light irradiation leads to an acceleration of the catalytic decomposition of methylene blue (MB).

## 2. Materials and Methods

The composite was prepared in the form of fibers by electrospinning using a 15 wt% solution of PVDF (Merck, Darmstadt, Germany) in a mixture of dimethyl sulfoxide and acetone with a volume ratio of 7:3. BFO was synthesized by the sol–gel method. To obtain PVDF/BFO, bismuth ferrite was dissolved in a mixing solvent in an amount of 5 wt% relative to the solid polymer before dissolving the PVDF [42]. Then, for 30 min, this solution was electrospun at a continuous 50 kV voltage. There was a 20 cm distance between the collector and the needle [43,44].

### 2.1. Characterization

A scanning electron microscope (SEM) Helios NanoLab 660 (ThermoFisher Scientific, Brno, the Czech Republic) was used to examine the morphology of the specimens. Leica EM ACE600 (Leica Microsystems, Wetzlar, Germany) coater was used to coat 15 nm of carbon to the samples.

Using an X-ray photoelectron spectrometer (XPS) AXIS SupraTM (Kratos Analytical Ltd., Manchester, UK), with an emission current of 15 mA, XPS spectra were captured. The software CasaXPS version 2.3.23 was used to fit the spectra.

Fourier-transform infrared spectroscopy (FTIR) measurements were performed using a Vertex70v (Bruker, Billerica, MA, USA) in transmission mode with 512 iterations and a 1 cm^−1^ resolution.

The spectra of the total transmittance $T_{t}$ and diffuse reflectance $R_{d}$ for the studied objects were measured in the wavelength range of (300 to 1000) nm using the Avasphere-50 integrating sphere (Avantes, Apeldoorn, the Netherlands). A combined deuterium/halogen lamp AvaLight-DH-S-BAL (Avantes, Apeldoorn, The Netherlands) was used as an illumination source; its radiation was supplied via 600 μm fiber-optic light guides. Photographic signals were registered using an automated spectrometer MS3504i (SOL-Instruments, Minsk, Belarus) coupled with a CCD matrix camera HS-101H-HR (Hamamatsu, Hamamatsu City, Japan). The final spectrophotometric coefficient data $T_{t}$ and $R_{d}$ were determined as: (1)$$R_{d}^{\exp}=\frac{R_{d}^{s}\;\left(\lambda \right)-R_{0}\;\left(\lambda \right)}{R_{\mathrm{gl}}\;\left(\lambda \right)-R_{0}\;\left(\lambda \right)}\;\mathrm{and}\;T_{t}^{\exp}=\frac{T_{t}^{s}\;\left(\lambda \right)-T_{0}\;\left(\lambda \right)}{T_{\mathrm{gl}}\;\left(\lambda \right)-T_{0}\;\left(\lambda \right)},$$
where $R_{d}^{s}\;\left(\lambda \right)$ and $T_{t}^{s}\;\left(\lambda \right)$ are the spectra of the transmission and reflection of samples; $R_{\mathrm{gl}}\;\left(\lambda \right)$ and $T_{\mathrm{gl}}\;\left(\lambda \right)$ are the spectra of the reference signal measured with the quartz plate; $R_{0}\;\left(\lambda \right)$ is the signal for a sphere with open optical ports; $T_{0}\;\left(\lambda \right)$ is the signal of the integrating sphere with covered input and open output ports.

The calculation of the spectral dependence of the optical absorption coefficient $\mu_{a}$ was carried out using the 2-flow Kubelka–Munk model [45,46].

Thermogravimetric and differential scanning calorimetry (TGA-DSC) curves were taken on a STA 449 F3 Jupiter (Netzsch, Selb, Germany) synchronous thermal analyzer at a heating rate of 10 °C/min in an argon atmosphere in alumina crucibles. Data processing and the integration of peaks were carried out using built-in application programs by “Netzsch”.

A vibrating-sample magnetometer (VSM) Cryogen-Free High Field Measurement System (Cryogenic, London, UK), which uses a linear magnetometer to vibrate a specimen fixed to a rod, was used to detect magnetic properties. Using a superconducting magnet and liquid helium, the magnetic field (up to 1 T) was generated. Near the detection coil, where the induced voltage was amplified and recorded with a capture time constant of 0.3 s, the sample oscillated at a frequency of 21 Hz. The measurements were performed with the field oriented parallel and perpendicular to the orientation of the PVDF fibers at 300 K room temperature.

PVDF/BFO film 3 × 1 cm${}^{2}$, which was used as an active piezoelectric component for the piezoelectric nanogenerator (PENG), was placed between the upper and lower Al foil electrodes. For compactness, the entire multi-layer structure was then thoroughly laminated with polypropylene (PP) adhesive tape. From both sides of the electrodes, conductive Al tapes were attached. The piezopotential was measured in two ways:The PENG was fixed on a flat surface and manually bent and unbent by 90°.The PENG was immersed in an ultrasonic bath, fixed stationary, and the ultrasound was turned on and off.

The output voltage from PENG was measured using a 2400-calibrator multimeter (Keithley, Cleveland, OH, USA).

### 2.2. Piezo-Photocatalytic Experiment

An ultrasonic bath with a power of 360 W and a frequency of 40 kHz was utilized for the piezo-photocatalytic decomposition. As UV and visible light sources, a 250 W high-pressure mercury lamp (Phillips) and a 70 W metal halide lamp (Osram) were chosen. The reactor was 10 cm away from the source of light. A temperature of no more than 30 °C was maintained continuously in the reactor. The test ran for 60 min, sampling every 15 min for the spectrophotometric analysis at a wavelength of 663.7nm.

A 3 × 1 cm${}^{2}$ sample weighing 5 mg was placed in a solution of 20 mL MB in a beaker. In order to distinguish between different factors, the decomposition of MB was the subject of additional investigations: under the influence of light with a catalyst (photocatalysis), under the influence of ultrasound without a catalyst (sonolysis), under the influence of ultrasound with a catalyst (piezocatalysis) and under the influence of light without a catalyst (photolysis).

## 3. Results

The optimization of the parameters for solution preparation was done according to previous work [47]. Scanning electron microscopy was used to describe the morphology of the prepared samples. The thickness of the mats was about 30 μm. Figure 1a,b shows SEM images of PVDF fibers modified with BFO nanoparticles. In Figure 1a, it is clear that the composite fibers have a preferential orientation and a non-uniform diameter. The histogram of the fiber diameter distribution in Figure 1c shows that the average diameter is in the order of 890 nm. Based on the fact that the surface of the fibers is rather uniform and smooth, it is evident that the BFO nanoparticles are incorporated within the fibers. The formation of homogeneous, dense, and defect-free fibers is associated with an increase in the charge density of the polymer solution [4]. Moreover, the formation of fibers depends on the space between the collector and the needle, the injection speed, and the applied voltage. It was shown by Mushtaq et al. [15] that with an increase in the distance between the needle and the collector and an increase in the injection speed, the applied electrospinning voltage must also increase to provide sufficient electrostatic forces to create a solution jet. To visualize the arrangement of nanoparticles, higher values of the current and accelerating voltage were used for the SEM analysis because these values are where the primary electron’s energy is enough to penetrate the barriers of the nanofibers and expose the internal structure. The results are shown in Figure 1b. It is apparent that the BFO nanoparticles are quite uniformly incorporated into the fibers. However, partial agglomeration is observed in certain areas. In addition, Figure 1b shows a slice of a single fiber, where it is clearly visible that the nanoparticles are located inside. As illustrated by the histogram of the nanoparticle diameter distribution in Figure 1d, the average particle size is about 250 nm.

To quantify the phase distribution of the pure and modified PVDF, Figure 2 shows the FTIR spectra in the range of (400 to 1600) cm${}^{-1}$.

The figure shows that the spectra do not differ significantly from one another. Peaks in both PVDF samples are observable at (614 and 763) cm${}^{-1}$, defining the non-polar α-phase. The peak locations at (445, 474, 1276, and 1431) cm${}^{-1}$ are associated with the piezoelectric β-phase. The polar γ-phase is characterized by peaks at (482, 812, and 1233) cm${}^{-1}$. The typical peaks are visible in all three α-, β-, and γ-phases, and are at (881, 1072, 1182, and 1403) cm${}^{-1}$ [48]. The relative abundance of each of the three α-, β-, and γ-phases in the samples were determined from the FTIR spectra. First, Formula (2) was used to obtain the fraction of the electroactive phase ($F_{\mathrm{EA}}$): (2)$$F_{\mathrm{EA}}=\frac{I_{\mathrm{EA}}}{\frac{K_{840*}}{K_{763}}I_{763}+I_{\mathrm{EA}}}\times 100\%,$$
where $I_{\mathrm{EA}}$ at 840* and $I_{763}$ at 763 cm^−1^ are the absorbencies; the absorption coefficients $K_{840*}$ and $K_{763}$ are at the corresponding wave numbers with values of (7.7 × 10${}^{4}$ and 6.1 × 10${}^{4}$) cm${}^{2}$ mol${}^{-1}$, respectively. Formulaes (3) and (4) were then used to determine the distributions of the β- and γ- electroactive phases: (3)$$F\left(\beta \right)=F_{\mathrm{EA}}\times \frac{{\Delta H}_{\beta^{\prime }}}{{\Delta H}_{\beta^{\prime }}+{\Delta H}_{\gamma }}\times 100\%,$$
and
(4)$$F\left(\gamma \right)=F_{\mathrm{EA}}\times \frac{{\Delta H}_{\gamma^{\prime }}}{{\Delta H}_{\beta^{\prime }}+{\Delta H}_{\gamma }}\times 100\%,$$
where ${\Delta H}_{\beta^{\prime }}$ and ${\Delta H}_{\gamma^{\prime }}$ are the absorbance (height) differences between the peak around at 1234 cm^−1^ and the nearest valley around at 1225 cm^−1^, and the peak at 1275 cm^−1^ and the nearest valley at 1260 cm^−1^, respectively.

Calculations revealed that the percentage of the α-phase in pure PVDF was 13.45%, the percentage of the γ-phase was 4.03%, and the percentage of the β-phase was 82.52%, respectively. After modification of BFO, the fraction of the γ-phase increased noticeably and amounted to 10.12%, the fraction of the α-phase was 6.85%, and the fraction of the β-phase underwent an insignificant increase to 83.03%. Based on the data obtained, it can be assumed that the introduction of BFO leads to phase repolarization due to the ion–dipole interaction, which could lead to local ordering of all trans-conformations. The effect of adding BFO on the polarization reversal of PVDF was discussed by Sasmal et al. [32], where it was demonstrated that the fraction of the polar phase increases from 38.2% for pure PVDF to 82.4% PVDF with the inclusion of 7 wt%
${\mathrm{Bi}}_{0.9}{\mathrm{Ba}}_{0.1}{\mathrm{FeO}}_{3}$. To study the optical characteristics of light absorption spectra in the range of (300 to 1000) nm were obtained. The results are shown in Figure 2b. Pure PVDF membranes, as illustrated, only exhibited UV light absorption in the 300 nm region. The content of BFO nanoparticles leads to an increase of $\mu_{a}$ in the fundamental absorption region with an extension of the absorption band to the region up to 450 nm. However, it should be noted that the PVDF/BFO is characterized by a sharp increase in the transport light scattering coefficient in the region above 500 nm. It is also important to note that in the region of basic absorption up to 450 nm, the transport scattering coefficient exceeds the absorption coefficient values by 4.5×.

The samples were examined by XPS in order to comprehend any potential interactions between the elements of the PVDF polymer and the nanoparticles. This technique has a sensitivity range of up to 5 nm. Pure PVDF and samples enhanced with BFO nanoparticles were both subjected to several XPS measurements. Detailed peak information was obtained by the Shirley background subtraction. Figure 3 shows the XPS spectra.

The high-resolution C, F, Bi, and Fe peak locations were determined after normalizing the spectra with the C1s position, which corresponds to the C-C bond (284.8 eV). In the overall spectra presented in Figure 3a, characteristic peaks for C, F, Bi, and Fe were identified. The absence of any other atoms indicates the purity of the sample within the error of the measurement method. To understand the features of the structure and mechanism of interfacial repolarization of the molecular surface chains of PVDF, one of the most interesting is the study of the state of the C1s high-resolution spectrum. From the data presented in Figure 3b, two characteristic regions can be distinguished for ${\mathrm{CF}}_{2}$ and ${\mathrm{CH}}_{2}$ dipoles.

Additionally, the C-C and C-O bond-related peaks may be seen at approximately 283.4 and 286.6 eV, respectively. For pure PVDF, the peak intensities are the same, while for PVDF fibers containing BFO, there are different peak ratios. It may also be observed from the high-resolution F1s spectrum in Figure 3c that the addition of nanoparticles leads to an increase in the peak intensity. All of this indicates structural rearrangements of the surface molecular chains of PVDF [49,50]. The presence of oxygen in pure PVDF is due to residual groups from solvents. From the overall spectrum and the high-resolution spectra for Bi4f and Fe2p (Figure 3d,e) it is apparent that the intensities of the peaks are very low. This is because the majority of the nanoparticles are placed deeper within the PVDF nanofibers than the analytical sensitivity limit of the XPS technique.

The thermal stability of the PVDF/BFO membrane was studied by DSC. Figure 4 shows the heating and cooling thermograms of the samples. The melting endotherm is characterized by a broad peak at 163.7 °C with a shoulder in the low-temperature region. This may indicate the presence of crystals of different sizes, crystal perfection, and the presence of different phases [51]. The crystallization isotherm indicates that the crystallization temperature ($T_{C}$) of the sample is 141.2 °C. Compared to pure PVDF [52] obtained by the same methodology, the crystallization temperature is higher by 10.8 °C. In addition, the TGA measurement was carried out simultaneously with the DSC. The results showed that the change in mass of the sample up to 190 °C is less than 1%. Thus, the results indicate high thermal stability of the obtained materials.

Hysteresis loops (M-H curves) were drawn in order to understand further the coercive force and saturation magnetization of the specimens. This was obtained by measuring the sample’s magnetization (M) as a function of the strength of the applied magnetic field (H) at a constant temperature. The magnetic field was applied parallel and perpendicular to the fiber orientation; the M-H magnetic hysteresis loops for the samples measured at 300 K are illustrated in Figure 5.

As revealed, the coercive force does not change with respect to the angle of inclination of the applied magnetic field, it is 212 Oe for both representations. The change occurs in the growth of magnetization, which tells us about the anisotropy of the magnetic properties of the composite itself. Separately, BFO powder and PVDF do not show such dependence. This is probably due to the fact that the addition of two materials creates an additional alignment of magnetic dipoles in one direction and may be due to both magnetocrystalline or magnetoelastic anisotropy or shape anisotropy.

Thus, the studies performed have demonstrated that the composite PVDF/BFO can be promising for use as a piezo-photocatalyst. The results of piezo-catalytic and piezo-photocatalytic measurements on the oxidation of the MB dye under the action of UV-visible and visible light are described in Figure 6. The measurement error usually did not exceed 5%.

Figure 6a,b displays the kinetic curves and the piezo-photocatalytic decomposition plots of MB under simultaneous sonication and UV-visible light. The photodegradation efficiency was 93% during 45 min. In a similar experiment with the absence of light (piezocatalysis), 64% decomposed over the same time. In the absence of both the light and a catalyst (sonolysis), it was 54%. The photocatalytic activity of 57% is similar to the photolysis of 58%, which may indicate the absence of photocatalytic properties in the composite since the diameter of the fibers is quite large and the nanoparticles are located inside a thick layer of the PVDF shell that prevents light penetration. The rate constants were also calculated using the pseudo-first-order kinetic equation, the values of which were 0.020; (0.019, 0.024, 0.065, and 0.020) min${}^{-1}$ for photolysis, sonolysis, piezocatalysis, and photocatalysis, respectively. The response rate has risen by (3.18, 3.29, 2.64, and 3.12)× under the simultaneous impact of UV light irradiation and ultrasonic treatment in comparison with photolysis, sonolysis, piezocatalysis, and photocatalysis, respectively. In piezo-photocatalytic experiments using visible light, the trend is similar. The highest activity is shown by piezo-photocatalysis, 83%, with piezocatalysis, 72%, and sonolysis, 62%. There is no significant difference between photocatalysis (23.8%) and photolysis (25%), which also confirms that the composite does not exhibit photocatalytic properties. From the calculated values of the rate constants (0.0048, 0.0176, 0.0229, 0.0281, and 0.0048) min${}^{-1}$ for photolysis, sonolysis, piezocatalysis, piezo-photocatalysis, and photocatalysis, it is evident that the speed of decomposition ramped up (5.9, 1.6, 1.2, and 5.9)× in comparison with photolysis, sonolysis, piezocatalysis, and photocatalysis, respectively.

When PVDF/BFO is exposed to US (piezocatalysis), there are two potential triggers for a set of catalyzed reactions:Forming a potential difference while being impacted by acoustic energy.A piezoelectric catalyst forms electron–hole pairs under the influence of shock waves produced by the burst of cavitation microbubbles.

Both of these processes can be represented by the equation: $\mathrm{PVDF}/\mathrm{BFO}+\mathrm{US}\to e^{-}+h^{+}$.

These surface charge states lead to redox reactions on the surface, leading to the generation of ·OH and $\cdot O_{2}^{-}$ radicals, which subsequently oxidize the dye molecules in the solution. Considering the depth of nanoparticles in the dielectric polymer matrix of fibers, as shown above, light irradiation does not lead to the excitation of charge states, and photocatalysis does not occur. However, when the composite was subjected to simultaneous light irradiation and ultrasonic vibration in addition to the piezocatalysis process mentioned above, the possibility of a piezo-photocatalytic process involving light, in this case, can be explained by the fact that the piezoelectric potential produced can increase the potential energy of electrons in the valence band, leading to band bending. In addition, the electronic energy bands can change significantly under the influence of ultrasonic exposure. As a result, the increased lifetime of the excited charges and the sloping band gap synergistically increase the catalytic efficiency. In our earlier study, it was demonstrated that the addition of iron oxide nanoparticles to PVDF nanofibers changes its electronic structure, creating local energy states near the Fermi level, leading to its photosensibilization [52]. Schematically it is represented in Figure 7. A similar effect was obtained by Zhang et al. [53], when ultrasonic irradiation led to photosensation of ${\mathrm{Na}}_{0.5}{\mathrm{Bi}}_{0.5}{\mathrm{TiO}}_{3}$, which initially did not show photocatalytic activity. Hence, more electron-hole pairs are formed in piezo-photocatalysis compared to piezocatalysis, resulting in a higher decomposition efficiency in MB decomposition.

To confirm the generation of the piezoelectric potential under mechanical action or ultrasonic treatment, a piezoelectric nanogenerator (PENG) was fabricated on the composite PVDF/BFO. PENG’s conceptual design and the stress signals that were acquired at a 90° bend and US load are shown in Figure 8a–c. The device (Figure 8a) is a simple multi-layer structure in which the PVDF/BFO film (3 × 1 cm${}^{2}$) used as an active piezoelectric component was placed between the upper and lower Al foil electrodes. For compactness, the entire multi-layer structure was then thoroughly laminated with polypropylene (PP) adhesive tape. From both sides of the electrodes, conductive Al taps were attached.

Figure 8b demonstrates the PVDF/BFO-based PENG time-dependent open-circuit stresses in tension and flexion at 90°. It is clear that the system produces open-circuit voltage values of about 60 mV that are adequately repeatable. Additionally, with ultrasonic irradiation of PVDF-based PVDF/BFO PENG, repeatable voltages (around 400 mV) were detected (Figure 8c). Furthermore, it was found that piezoelectric charges in PVDF/BFO may be constantly stimulated by both mechanical bending and ultrasonic waves. Such continuous stimulation results in an alternate internal electric field that can efficiently transfer charges to the PVDF/BFO-solution interface.

## 4. Conclusions

As a result of the study, it can be assumed that the use of ${\mathrm{BiFeO}}_{3}$ as a filler causes the electroactive phase to grow, which leads to an increase in the efficiency of piezocatalysis. The composite material does not exhibit photocatalytic activity; however, under the combined action of light and US, the rate of the MB decomposition reaction increases by about 3× under UV irradiation and by about 6× under visible irradiation. Such an acceleration of the reaction can be explained by the presence of the piezo-photon effect, in which the generated piezoelectric potential leads to band bending and activation of the photocatalytic activity of the composite. The presence of magnetic anisotropy may then be utilized as an additional parameter to modulate photocatalytic characteristics. 

## Figures and Tables

**Figure 1 polymers-15-00246-f001:**
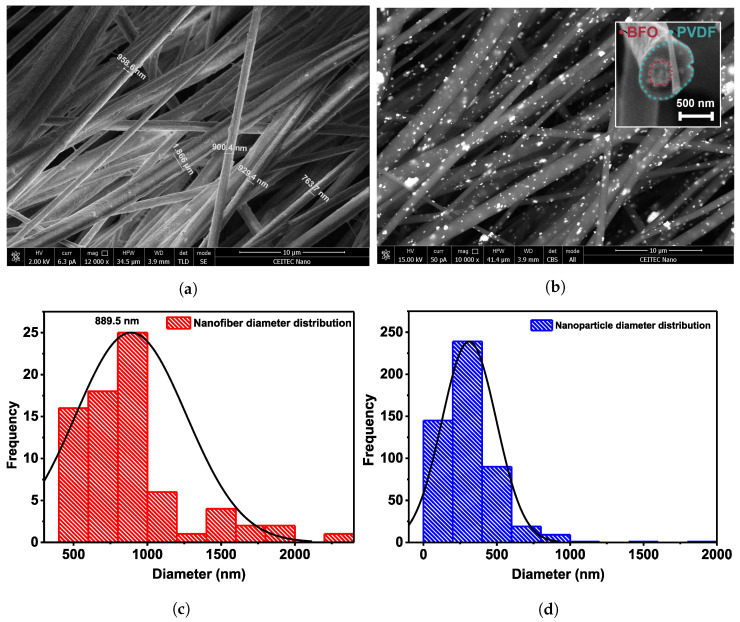
(**a**,**b**) SEM images of PVDF/BFO fibers. (**c**) Nanofiber diameter distribution and (**d**) nanoparticle size distribution diagrams.

**Figure 2 polymers-15-00246-f002:**
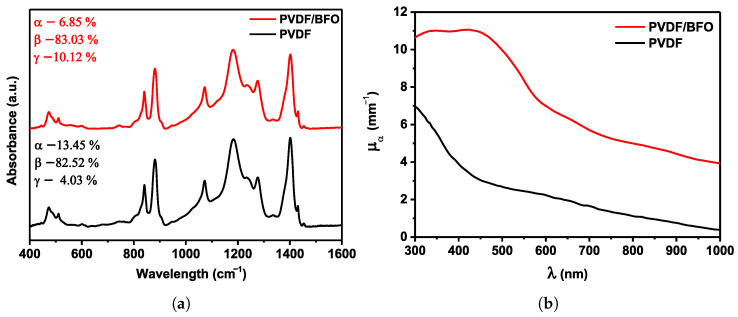
(**a**) FTIR absorption spectra of fibers made of PVDF/BFO and pure PVDF. (**b**) Absorption coefficient spectra—$\mu_{a}$ of PVDF/BFO and pure PVDF.

**Figure 3 polymers-15-00246-f003:**
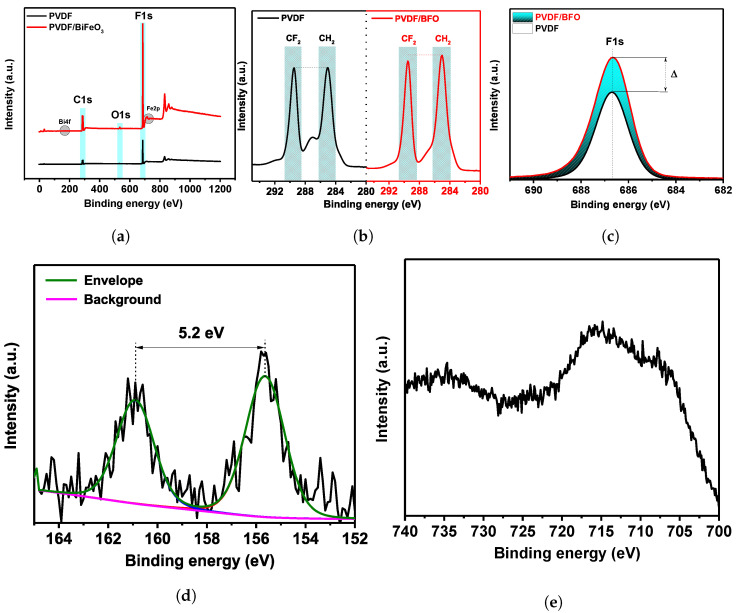
XPS PVDF and PVDF/BFO spectra: (**a**) wide spectrum, (**b**) high-resolution C1s spectra, (**c**) high-resolution F1s spectra, (**d**) high-resolution Bi4f spectra, (**e**) high-resolution Fe2p spectra.

**Figure 4 polymers-15-00246-f004:**
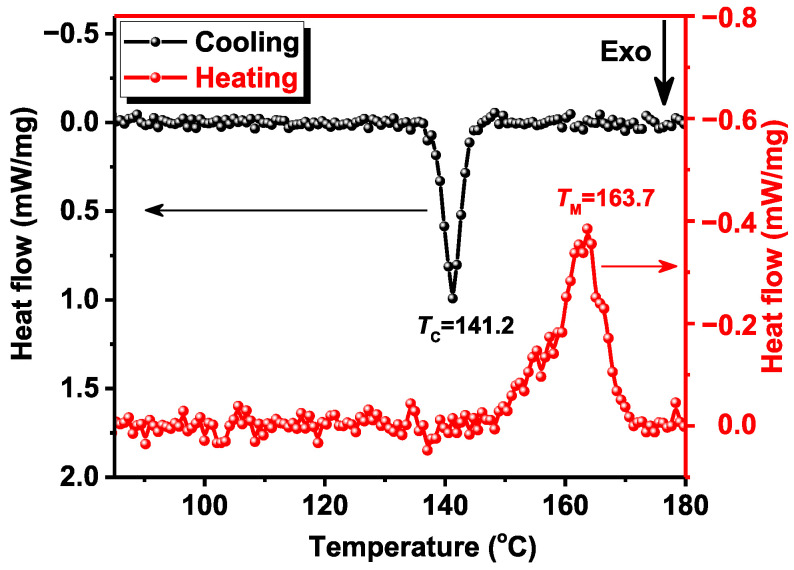
DSC thermographs of PVDF/BFO fibers.

**Figure 5 polymers-15-00246-f005:**
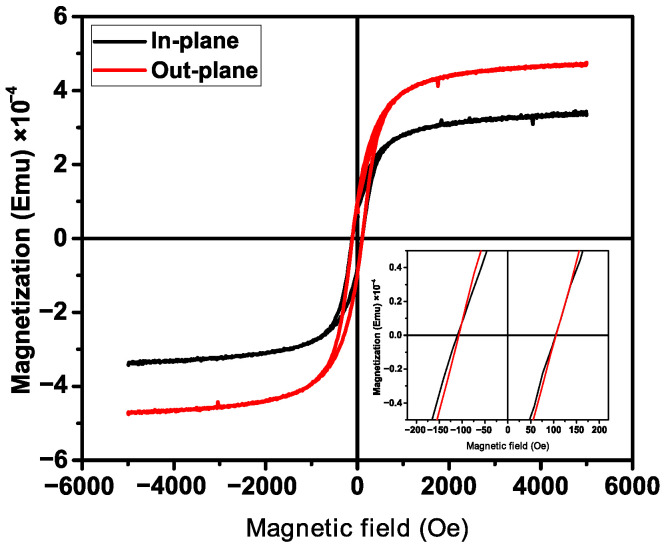
Room temperature M-H loops for PVDF/BFO, measured under in-plane and out-plane conditions at 300 K.

**Figure 6 polymers-15-00246-f006:**
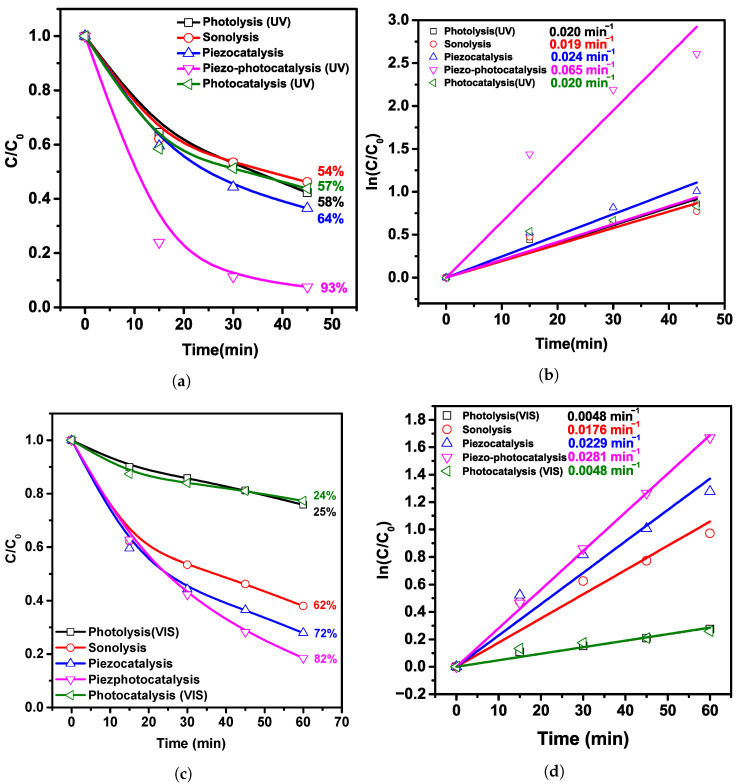
(**a**) MB degradation curves (1 mg L^−1^, 20 mL) and (**b**) time dependence of ln(*C*/$C_{0}$) for PVDF/BFO nanofibers under UV irradiation; (**c**) MB degradation curves (1 mg L^−1^, 20 mL) and (**d**) time dependence of ln(*C*/$C_{0}$) for PVDF/BFO nanofibers under visible light.

**Figure 7 polymers-15-00246-f007:**
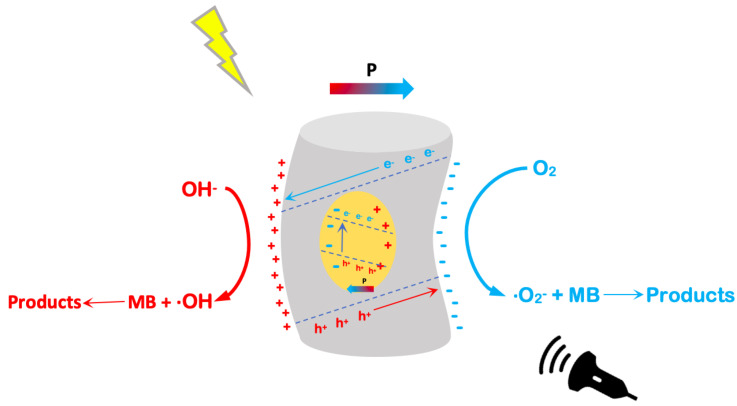
Piezo-photocatalysis of PVDF/BFO under light irradiation and ultrasonic vibration: proposed operating mechanism.

**Figure 8 polymers-15-00246-f008:**
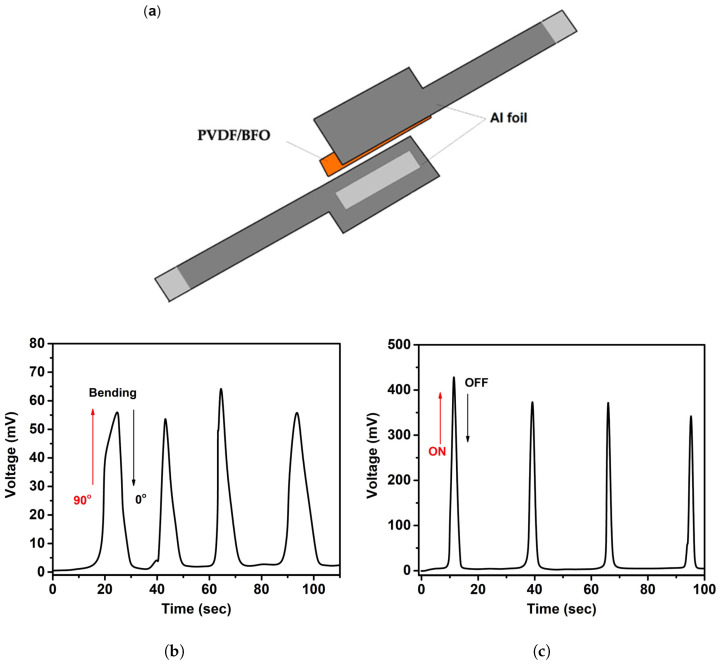
(**a**) Schematic structure of the PVDF/BFO-based PENG. (**b**) The PVDF/BFO-based PENG time-dependent open-circuit voltages under 90° bending and (**c**) US stress.

## Data Availability

Data will be provided upon personal request from Farid Orudzhev (farid-stkha@mail.ru) and Dinara Sobola (sobola@vut.cz).

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
