# Peer review of "Piezo-Enhanced Photocatalytic Activity of the Electrospun Fibrous Magnetic PVDF/BiFeO3 Membrane"

_polymers, 2023, doi:10.3390/polym15010246_

Round 1

Reviewer 1 Report

This paper synthesized a hybrid polymer-inorganic PVDF/BFO fibrous membrane using electrospinning for piezo-photo catalytic decomposition. Due to the synergic effect between BFO nanoparticles and PVDF polymeric matrix, the fiber composites exhibit a specific piezo-photo-catalytic performance. And the composites are indeed capable of inducing oxidation of MB dye under the action of UV-visible and visible light. This manuscript reports good results, but I believe that the following points need to be improved.

1. The Introduction part, some recently published papers about piezo-/photo-composite fibers fabricated through electrospinning should be mentioned to enrich the significance and background of the present work. such as:

https://doi.org/10.1088/1361-6528/ac7ed5

https://doi.org/10.1007/s12221-022-4099-y

2. In line 114-116, the authors state that “Based on the fact that the surface of the fibers is rather uniform and smooth, it can be concluded that the BFO nanoparticles are located inside the fibers.” However, I can not agree with this conclusion just according to the SEM observation. I suggest the authors to perform the TEM observation to make an accurate conclusion.

3. I recommend the authors to give more details about the piezoelectric performance test, such as the amplitude of applied force, frequency, and how to applied these external mechanical stimulus.

Author Response

The responses can be found in the attached file.

Reviewer 2 Report

Farid Orudzhev et al reported PVDF/BFO composites fibers that can work as the piezo-photo-catalyst to improve MB degradation.  This is an interesting topic, however, the authors must carefully revise their manuscript before it can be published in Polymers.

1) What is the meaning of " It is shown that the addition of 6 BPO leads to an increase in the proportion of the polar phase from 86.5 ◦C to 96.1 ◦C due to the surface 7 ion-dipole interaction."? The main text does not mention these values, and is it oC?

2) The UV-vis absorbance spectra of the PVDF/BFO fibers should be offered to help readers understand why visible light and UV light work so differently.

3) Errors bars should be given for the data in Figure 5(c) to convince readers that the light irradiation substantially improve catalysis.

4) Is there any relevance between the figure 4 and the major contribution of those work?

5) The authors must carefully check their writing, many typos.

Author Response

(The authors gave the same response as above.)

Reviewer 3 Report

Manuscript (MS) polymers-2010117 entitled “Piezo-enhanced photocatalytic activity of electrospun fibrous magnetic PVDF/BiFeO3 membrane reports the formulation, synthesis, characterization and piezo-photocatalytic (even piezophoton effect) of original electrospun magnetic PVDF/BiFeO3 composites. BFO nanoparticles are embedded inside PVDF, well identified by SEM images. These NPs enable to enhance the electroactive properties of PVDF.

This is a nice piece of work well supported by careful characterizations (e.g. the ferroelectric b-phase well observed by IR spectra).

The overall feelings are that this well-presented and illustrated MS is a bit short and should be improved by additional data (especially thermal ones) see my comments below, though it falls in the scope of the journal. Indeed, there are further issues to address to be this MS accepted in Polymers, below:

Several Comments:

1. Regarding the choice of solvents (DMSO/acetone) for electrospinning, I assume that optimization has been done and one comment on this would be appropriate. Please insert ina small paragraph at the beginning of the “3. Results”.

2. How thick was(were) the mat(s)? How many done?

3. The authors have not supplied any DSC thermograms to ensure they observed a possible Curie temperature. Also what about the thermal stability?

4. Though the literature is fine, several key references are missing: i) for electrospun of F-polymers, may I suggest to insert the following reveiw: Polym Rev. 2022 (10.1080/15583724.2022.2067868). ii) In addition, an excellent review on electroactive phases of PVDF (as well as their determination, processing and applications) Prog Polym Sci 2014, 39, 683-706 also deserves to be included; iii) two book chapters to cite in the introduction: Altomare A, Bozorg M, Loos K. PVDF-Based Multiferroic. In: Ameduri B, Fomin S, editors. Fluoropolymers: Research, Production Issues, and New Applications. Fascinating Fluoropolymers and Applications, Progress in Fluorine Science, vol. 2. Amsterdam: Elsevier. 2020. p. 45-81 and Costa CM, Fernandes Cardoso V, Brito-Pereira R, Martins P, Correia DM, Correia V, Ribeiro C, Martins PM, Lanceros-Mendez S. Electroactive Poly(vinylidene fluoride)-based Materials: Recent Progress, Challenges, and Opportunities. In: Ameduri B, Fomin S, editors. Fluoropolymers: Research, Production Issues, and New Applications. Fascinating Fluoropolymers and Applications, Progress in Fluorine Science, vol. 2. Amsterdam: Elsevier. 2020. Chapt. 1. p. 1-43;

5. In line 180 (page 6) the authors state that NPs are located within the polymer; I suggest that SEM of the slice of the mat be analyzed to confirm that feature.

6. It is a pity that XRD spectra have been supplied.

Typo: - page6 lines 174-175 redundancy of “it can be seen”

line 189 is 117Oe correct?

Author Response

(The authors gave the same response as above.)

Round 2

Reviewer 2 Report

The authors adequately addressed my concern and it can be published in Polymers.